# *Cryptosporidium* spp.: Human incidence, molecular characterization and associated exposures in Québec, Canada (2016-2017)

**Juliana Ayres Hutter[1], Réjean Dion[2,3], Alejandra Irace-Cima[1], Marc Fiset[4], Rebecca Guy[5], Brent Dixon[6], Jeannie Lisette Aguilar[2], Julien Trépanier[2], Karine Thivierge[2,7]***

**1** Direction des risques biologiques et de la santé au travail, Institut national de santé publique du Québec, Montréal, Québec, Canada, **2** Laboratoire de santé publique du Québec, Institut national de santé publique du Québec, Sainte-Anne-de-Bellevue, Québec, Canada, **3** Département de médecine sociale et préventive, École de santé publique de l'Université de Montréal, Montréal, Québec, Canada, **4** Direction de la vigie sanitaire, Ministère de la Santé et des Services sociaux, Québec, Québec, Canada, **5** National Microbiology Laboratory at Guelph, Public Health Agency of Canada, Guelph, Ontario, Canada, **6** Bureau of Microbial Hazards, Food Directorate, Health Canada, Ottawa, Ontario, Canada, **7** Institute of Parasitology, Faculty of Agricultural and Environmental Sciences, McGill University, Macdonald Campus, Sainte-Anne-de-Bellevue, Québec, Canada

* karine.thivierge@inspq.qc.ca

**Data Availability Statement:** All relevant data are within the manuscript.

## Abstract

The aim of this study was to describe the epidemiology of human cryptosporidiosis in Québec from 2016 to 2017 and to identify possible exposures associated with the disease, and the dominant *Cryptosporidium* species in circulation. A descriptive analysis was performed on data collected from the provincial notifiable infectious diseases registry and the epidemiological investigation. Fecal sample were sent to the Laboratoire de santé publique du Québec for molecular characterization. In Québec, from January 1, 2016 to December 31, 2017, a total of 201 confirmed cases of cryptosporidiosis were notified. A peak in the number of reported cases was observed at the end of the summer. The regional public health department with the highest adjusted incidence rate for sex and age group for both years was that of Nunavik, in the north of Québec. A higher average annual incidence rate was observed for females between the ages of 20 to 34 years compared to males. Overall, for both males and females the distribution appeared to be bimodal with a first peak in children younger than five years old and a second peak in adults from 20 to 30 years of age. Molecular characterization showed that 23% (11/47) of cases were infected with *C. hominis* while 74% (35/47) were infected with *C. parvum*. Meanwhile, subtyping results identified by gp60 sequencing, show that all *C. parvum* subtypes belonged to the IIa family, whereas the subtypes for *C. hominis* belonged to the Ia, Ib, and Id families. Finally, the epidemiological investigation showed that diarrhea was the most common reported symptom with 99% (72/73) of investigated cases having experienced it. This first brief epidemiological portrait of cryptosporidiosis in Québec has allowed for the description, both at the provincial and regional level, of the populations that could be particularly vulnerable to the disease.

**Funding:** Work related to this study was supported by the Institut national de santé publique du Québec (KT, JAH).

**Competing interests:** The authors have declared that no competing interests exist.

## Introduction

*Cryptosporidium* spp. are among the most frequently reported pathogenic enteric protozoans in North America and elsewhere in the world [1]. Cryptosporidiosis can cause severe diarrhea in young children and can be fatal for immunosuppressed people [2–4]. Infective oocysts are highly resistant in the environment and are usually ingested either directly through the fecal-oral route from infected hosts (human or animal) or indirectly through water or food that has been contaminated by fecal matter [2,3]. Two species cause the majority of infections in humans, *Cryptosporidium hominis* and *Cryptosporidium parvum*. These two species have different reservoirs, transmission modes and hosts. *Cryptosporidium hominis* is mainly transmitted between humans whereas *C. parvum* has been associated with transmission between humans and animals [2–4].

In North America, cryptosporidiosis is one of the leading causes of waterborne outbreaks [5,6]. In the Unites States, between 1991 and 2002, 7% of drinking water related outbreaks were attributed to *Cryptosporidium* [7], whereas for 2013 to 2014, close to 28% of drinking water related outbreaks were attributed to it [8]. It has been estimated that 748,000 cases of cryptosporidiosis occur each year in the United States with an annual cost of US$45.8 million in hospitalisations, while only 2% of these cases are reported [5]. Likewise, in Canada, between 1974 and 2001, *Cryptosporidium* was the third most associated pathogen with drinking water outbreaks [9]. Out of the 238 disease outbreaks for which etiological agents were identified, 12 were attributed to *Cryptosporidium* [9]. In Canada, cryptosporidiosis became a notifiable disease in the year 2000 [10]. Since 2000, the number of reported cases has been approximately 600 per year, however a slight increase can be seen from 2011. In 2001, a drinking water related outbreak in North Battleford, Saskatchewan, saw more than 1,700 cases reported for that year [11]. In Québec, cryptosporidiosis became a provincially notifiable disease in 2003. The number of reported cases has varied from around 30 in 2004 to more than 120 cases in 2015, when an outbreak was declared in the northern region of Nunavik [12,13]. It is highly likely that, as in the United States, the number of reported cases of cryptosporidiosis in Québec, and in Canada, represent a fraction of the number of actual cases [14].

The substantial underdiagnosis of the disease coupled with the absence of systematic investigation of cases by the regional public health departments (RPHDs), results in insufficient data and knowledge of the epidemiology of the disease in Québec. In this context, the overall aim of this study was to describe the epidemiology of human cryptosporidiosis in Québec from 2016 to 2017. More precisely, the present study aimed to identify possible exposures associated with the disease, and the dominant *Cryptosporidium* species in circulation in Québec.

## Materials and methods

### Data

The collection, use, analysis, and disclosure of data described here fall within an enhanced surveillance mandate from the Ministère de la Santé et des Services sociaux (MSSS) with the accord from the RPHDs. Therefore, research ethics committee approval was not required. The data for the descriptive analyses come from two different sources: the notifiable infectious diseases registry (NIDR), which is the passive surveillance database of the RPHD and MSSS, and from the epidemiological investigation of cryptosporidiosis cases carried out on a voluntary basis by the RPHD. Cases are counted in the NIDR by their health region of residence (HRR). All data used in the current study were anonymized before their reception.

For the first part of the descriptive analysis, all notified cases of cryptosporidiosis in Québec from January 1st 2016 to December 31st 2017 were extracted from the NIDR on August 6th 2018,

according to their dates of notification reception (episode date). Only confirmed cases were included in the study. Confirmed cryptosporidiosis cases are defined as either the presence of *Cryptosporidium* oocsts in a fecal specimen, detection of *Cryptosporidium* DNA in a fecal specimen, or detection of antigens by enzyme immunoassay or direct immunofluorescence [15] Extracted data were imported and transformed with the EpiData Entry 3.1 software (URL: https://www.epidata.dk/). The following variables were included in the extraction: age, sex, HRR, episode date, onset of illness date, clinical evolution of the disease, specimen collection date, nature of specimen and laboratory analysis result. The episode date was used for the time dimension of analysis of the data.

For the second part of the descriptive analysis, all 18 RPHDs in Québec were solicited to carry out an epidemiological investigation of cryptosporidiosis cases from January 1st 2016 to December 31st 2017. A standard questionnaire was developed by the Laboratoire de santé publique du Québec (LSPQ) of the Institut national de santé publique (INSPQ) and the MSSS for this purpose. The questionnaires were then sent back to the INSPQ for data compilation and analysis. In Québec, the RPHD do not usually investigate all cryptosporidiosis cases. The period of exposure for acquisition of the disease was defined as the period of time beginning two weeks prior and up to the date of onset of illness; this interval was used to cover the minimal and maximal boundaries of the known incubation period for cryptosporidiosis of 1 to 12 days.

## Collection and molecular analysis of fecal specimens

In May 2016, all medical microbiology laboratories across Québec were asked to send to the LSPQ frozen unfixed stool samples from patients with a laboratory confirmed *Cryptosporidium* spp. infection. All samples were from patients diagnosed from January 1st 2016 to December 31st 2017. Following collection, all samples were stored at -70˚C.

Species and genotypes of *Cryptosporidium* isolates were determined by nested-PCR amplification and sequencing of a portion of the gene encoding the small subunit (SSU) rRNA, according to Nichols et al. [16]. For further genotyping, a 450 bp fragment of the 60 kDa glycoprotein (gp60) gene was amplified according to the protocol described by Iqbal et al. [17]. Genetic subtyping was determined by DNA sequence analysis of the 60 kDa glycoprotein (gp60) gene. Each sequence for all the gene fragments was independently compared to Gen-Bank and CryptoDB sequences of *Cryptosporidium* species by BLAST analysis [18]. Sequences were aligned using BioNumerics version 7.6.3 (Austin, TX, USA).

## Statistical analysis

Frequencies and proportions were calculated for all reported cases from the NIDR by age, sex and HRR. Incidence rates (IR) were calculated for the 2016 and 2017 years using the number of new reported cases as the numerator, and the population of Québec per year as the denominator. Age and sex IR standardization were done with the direct method, with the Québec population as the reference so the IR from different regions could be compared. Mean annual incidence rates were calculated by using the total number of cases from both 2016 and 2017 as the numerator and the population of Québec for both years as the denominator. Frequencies and proportions were also calculated for the data from the epidemiological investigation according to variables of interest such as symptomology and exposure factors for the transmission of *Cryptosporidium*. In view of the small number of data points obtained from the epidemiological investigation, the data from both years, 2016 and 2017, were pooled together. Missing data were excluded from analyses. Statistical analyses were done with the software SAS/STAT version 9.4 (SAS Institute Inc., Cary, NC, USA).

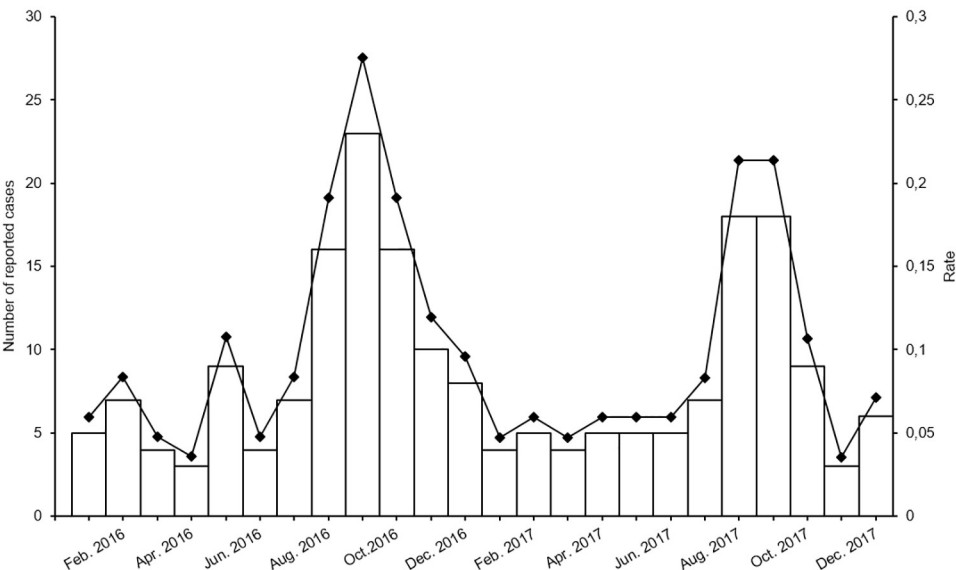

**Fig 1. Number and crude incidence rate of reported cases by month of notification, 2016–2017.** Crude incidence rate per 100,000 population.

## Results

### Descriptive analyses of reported cases

In Québec, from January 1, 2016 to December 31, 2017, a total of 201 confirmed cases of cryptosporidiosis were registered in the NIDR. For 2016, there were 112 cases reported (1.3 per 100,000 population), while for 2017 there were 89 cases reported (1.1 per 100,000 population). Fig 1 shows the number of cases by month of notification reception. A peak in the number of reported cases is noticeable at the end of the summer, while the lowest numbers of reported cases are during winter and spring.

Table 1 shows the number, percentage and age and sex-adjusted incidence rate by HRR of cases for 2016 and 2017. Out of the 18 HRR, only 12 had reported cases in 2016 and 2017. The adjusted incidence rate ranged from 0.35 cases per 100,000 population in Montérégie to 15.05 cases in Nunavik for 2016, and from 0.12 in Capitale-Nationale to 7.78 in Nunavik for 2017. It is important to keep in mind that incidence rates resulting from a small number of cases and population tend to have poor reliability.

For both years, females represented 55% of all reported cases. Fig 2 shows the average annual crude incidence rate by sex and age group. A higher average annual incidence rate can be seen for females between the ages of 20 to 34 years, inclusive, whereas a lower incidence rate can be seen for females between 5 to 14 years, inclusive. Overall, for both males and females the distribution appears to be bimodal with a first peak in children younger than five years old and a second peak in adults from 20 to 30 years of age.

### Descriptive analyses according to *Cryptosporidium* species and subtype of reported cases

Stool samples were sent to the LSPQ for genotyping for 35% of reported cases (70/201) during the study period. Genotype and subtype sequencing was successful for 67% (47/70) and 50% (35/70) of samples, respectively. Genotyping showed that 23% (11/47) of these cases were infected with *C. hominis* while 74% (35/47) were infected with *C. parvum*. Only a single case

**Table 1. Number, percentage and sex and age-adjusted incidence rate of confirmed reported cases by health region of residence (HRR) in Québec, Canada, 2016–2017.**

| HRR | 2016 | | 2017 | |
|---|---|---|---|---|
| | *n* (%) | Incidence rate [a] | *n* (%) | Incidence rate [a] |
| Bas-Saint-Laurent | - | - | 2 (2.2) | 1.17 |
| Capitale-Nationale | 4 (3.6) | 0.53 | 1 (1.1) | 0.12 |
| Estrie | 13 (11.6) | 2.85 | 4 (4.5) | 0.90 |
| Montréal | 13 (11.6) | 0.58 | 16 (18.0) | 0.70 |
| Outaouais | 4 (3.6) | 0.73 | 4 (4.5) | 0.98 |
| Abitibi-Témiscamingue | 3 (2.7) | 2.00 | 4 (4.5) | 2.69 |
| Chaudière-Appalaches | 38 (33.9) | 9.43 | 28 (31.5) | 6.96 |
| Laval | 2 (1.8) | 0.41 | 1 (1.1) | 0.22 |
| Lanaudière | 7 (6.3) | 1.34 | 4 (4.5) | 0.78 |
| Laurentides | 21 (18.8) | 3.73 | 17 (19.1) | 2.93 |
| Montérégie | 6 (5.4) | 0.35 | 7 (7.9) | 0.49 |
| Nunavik | 1 (0.9) | 15.05 | 1 (1.1) | 7.78 |

[a] Age and sex-adjusted incidence rate for 100,000 population

was infected with *C. ubiquitum* (2%). One case infected with *C. hominis* lived in Vancouver and was thus excluded from further analyses. Overall, for both study years combined, females represented 80% (8/10) of *C. hominis* cases while males represented 66% (23/35) of *C. parvum* cases. Table 2 shows the number and proportion of cases according to genotype and demographic characteristics. For *C. parvum* cases, the 0 to 14 years age group had the highest proportion of cases in 2016, while in 2017 it was the 15 to 29 years age group. For *C. hominis* cases, the 0 to 14 years age group had the highest proportion of cases for both years. In regards to the HRR, most of the samples came from the Chaudière-Appalaches region for both *Cryptosporidium* species.

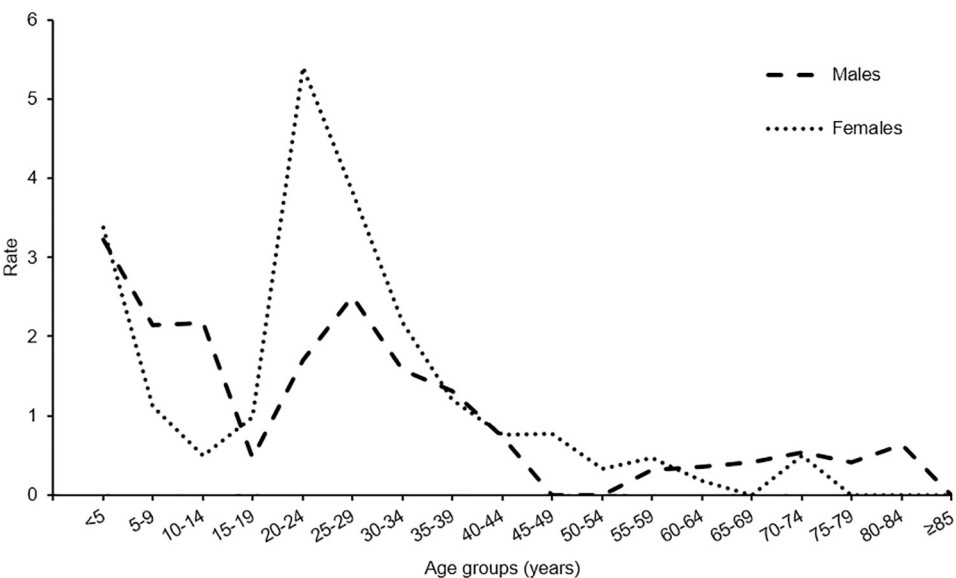

**Fig 2. Number and average annual crude incidence rate of cases by age group and sex, 2016–2017.** Average annual crude incidence rate for 100,000 population.

**Table 2. Number and percentage of confirmed reported cases by *Cryptosporidium* species in Québec, Canada, 2016–2017.**

| Variable | *C. parvum n* (%) | | *C. hominis n* (%) | |
|---|---|---|---|---|
| | **2016** | **2017** | **2016** | **2017** |
| **Sex** | | | | |
| Male | 16 (73) | 7 (54) | - | 2 (33) |
| Female | 6 (27) | 6 (46) | 4 (100) | 4 (67) |
| **Age (years)** | | | | |
| <15 | 9 (41) | 2 (15) | 2 (50) | 3 (50) |
| 15–29 | 7 (32) | 5 (38) | - | 2 (33) |
| 30–44 | 5 (23) | 4 (31) | 1 (25) | 1 (17) |
| 45–59 | - | 1 (8) | 1 (25) | - |
| 60–74 | - | - | - | - |
| 75–84 | - | - | - | - |
| ≥85 | 1 (5) | - | - | - |
| Missing | - | 1 (8) | - | - |
| **HRR** | | | | |
| Capitale-Nationale | 1 (5) | - | - | - |
| Mauricie et Centre-du-Québec | - | 1 (8) | - | - |
| Estrie | 1 (5) | - | - | - |
| Nord-du-Québec | - | 1 (8) | - | - |
| Chaudière-Appalaches | 17 (77) | 9 (69) | 1 (25) | 4 (67) |
| Laurentides | 1 (5) | - | - | - |
| Montérégie | 1 (5) | 1 (8) | 3 (75) | 2 (33) |
| Missing | 1 (5) | 1 (8) | - | - |
| Total | 22 (100) | 13 (100) | 4 (100) | 6 (100) |

Subtyping results are shown in Table 3. For all subtypes identified by gp60 sequencing, all *C. parvum* subtypes belonged to the IIa family, whereas the subtypes for *C. hominis* belonged to the Ia, Ib, and Id families.

## Descriptive analyses of the epidemiological investigation data

An epidemiological investigation was undertaken for 36% (73/201) of all reported cases in Québec during the study period. In 2016, 40 (55%) cases were investigated, while in 2017, 29

**Table 3. Number and percentage of confirmed reported cases by *Cryptosporidium* subtype in Québec, Canada, 2016–2017.**

| Subtype | *n* (%) |
|---|---|
| *C. hominis* IaA22R3 | 1 (3) |
| *C. hominis* IbA9G3 | 1 (3) |
| *C. hominis* IbA10G2 | 6 (17) |
| *C. hominis* IdA17G1 | 2 (6) |
| *C. hominis* IdA19 | 1 (3) |
| *C. parvum* IIaA15G2R1 | 13 (37) |
| *C. parvum* IIaA16G3R1 | 4 (11) |
| *C. parvum* IIaA16G2R1 | 3 (9) |
| *C. parvum* IIaA17G2R1 | 2 (6) |
| *C. parvum* IIAa17G3R1 | 2 (6) |
| Total | 35 (100) |

**Table 4. Number and percentage of investigated cases by health region of residence (HRR) in Québec, Canada, 2016–2017.**

| HRR | *n* (%) |
|---|---|
| Capitale-Nationale | 4 (5) |
| Estrie | 6 (8) |
| Abitibi-Témiscamingue | 7 (10) |
| Chaudière-Appalaches | 12 (16) |
| Laval | 2 (3) |
| Lanaudière | 9 (12) |
| Laurentides | 30 (41) |
| Montérégie | 3 (4) |
| Total | 73 (100) |

(40%) cases were investigated. Four cases (5%) had no episode dates, but their symptom onset dates were in 2016. Females represented 48% (35/73) of cases and males 52% (38/73). The mean age of investigated cases was 21 years (range 1 to 77 years), with a median age of 23 years. The age distribution of investigated cases was very similar to that of reported cases, with 75% of these being under 30 years. A bimodal distribution with a first peak in children younger than 10 years, and a second in young adults between 25 and 35 years of age, was also seen for investigated cases. In Table 4, the distribution of cases is shown according to HRR of investigated cases. The Laurentides health region had the highest proportion of investigated cases (41%).

Diarrhea was the most common reported symptom with 99% (72/73) of investigated cases having experienced it (Table 5). The other most common reported symptoms by investigated cases were: abdominal cramps (74%), appetite loss (58%) and fatigue (49%). Hospitalizations occurred in 15% (11/73) of investigated cases. The mean age of hospitalized cases was 26 years of age with a range from 1 to 63 years of age.

Exposures associated with the transmission of cryptosporidiosis were included in the investigation questionnaire. These factors included recreational water activities, raw food consumption, animal contact, at risk human contact, and other high risk environments attendance. Each exposure was divided into subcategories. Recreational water exposure was divided into beach swimming, pool swimming, and swimming at a water park. Among investigated cases, 19% (14/73) went swimming at the beach during the exposure period, while 25% (18/73)

**Table 5. Number and percentage of investigated cases by reported symptoms in Québec, Canada, 2016–2017.**

| Symptoms | *n* (%) |
|---|---|
| Diarrhea | 72 (99) |
| Nausea | 25 (34) |
| Vomiting | 27 (37) |
| Abdominal cramps | 54 (74) |
| Fatigue | 36 (49) |
| Appetite loss | 42 (58) |
| Dehydration | 17 (23) |
| Fever | 27 (37) |
| Weight loss | 25 (34) |
| Other clinical manifestations | 11 (15) |
| Total | 73 (100) |

swam in a pool, and 5% (4/73) swam at a water park. Raw food consumption was divided into raw fruits, raw vegetables, and unpasteurized drinks, such as fresh fruit juice and milk. During the exposure period, 11% (8/73) of investigated cases consumed raw fruits, 12% (9/73) consumed raw vegetables and 8% (6/73) consumed unpasteurized drinks. Only one case (1%) consumed fresh raw herbs during the exposure period, and one case (1%) consumed raw shellfish. Animal contact exposure was divided into companion animals, farm animals, zoo animals, and "others". Among investigated cases, 62% (45/73) had contact with companion animals during the exposure period, while 29% (21/73) had contact with farm, zoo or other animals. Human contact was divided into three categories: contact with a person with diarrhea, contact with a child in diapers, and contact with a child attending a daycare center. During the exposure period, 38% (28/73) of cases had contact with a person with diarrhea, 27% (20/73) had contact with a child in diapers, and 33% (26/73) had contact with a child attending a daycare center. As for attendance of high risk environments, 25% (18/73) of cases attended a daycare center during the exposure period. The mean age for the latter group was 2.5 years of age with a range of 1 to 6 years of age.

## Discussion

To our knowledge, this is the first provincial study attempting to characterize the epidemiology and exposures associated with cryptosporidiosis across all age groups and regions in Québec. The crude incidence rates for the years 2016 and 2017 were lower than 2015 ($IR_{2015}$ = 1.53) [12], which saw a large outbreak in the arctic region of Nunavik, in Québec [13]. However, since 2013 the incidence rate of cryptosporidiosis has been on the rise and appears to be higher than that of the 2004 ($IR_{2004}$ = 0.48) to 2012 ($IR_{2012}$ = 0.47) period (unpublished manuscript). This increase could be explained by a few potential reasons such as the implementation of new laboratory tests, better awareness of the disease by health practitioners (i.e., an increase in the number of tests ordered), an increase in the number of outbreaks and cases, or a combination of all these factors. The monthly distribution of the disease suggests that cryptosporidiosis is present throughout the year with peaks in late summer. Studies from Canada [19,20], the United States [5] and Europe [21] show a similar pattern. The upsurge of cases towards the end of the summer has been associated with an increase in recreational water activities [5,22,23]. In this study 16% of cases had been exposed to recreational water. The IbA10G2 subtype was the predominant *C. hominis* subtype in this study as is worldwide [24] and is the predominant subtype linked with recreational and drinking water outbreaks in the UK [25]. In addition, the *C. hominis* IdA19 subtype seen in this study has been linked to cases from a community recreational centre in British Colombia, Canada, and along with IbA10G2 was associated with the large drinking water outbreak in Saskatchewan, Canada [26].

For both years of the study, the Chaudière-Appalaches region has the highest adjusted incidence rate of cryptosporidiosis, followed by the regions of Estrie and Laurentides. Differences in services offered, as well as available technology among the regions, may explain to some extent the different incidence rates observed. In fact, without a specific laboratory requisition, the detection of *Cryptosporidium* oocysts is not part of the conventional ova and parasites (O&P) exam, which is a contributing factor to the underdiagnosis of cryptosporidiosis in Québec. On the other hand, some laboratories use a multiplex PCR for the diagnosis of intestinal protozoa, instead of the conventional O&P, which allows for the detection of *Cryptosporidium* spp. DNA in fecal specimens. This allows for better rates of diagnosis as it doesn't require a specific *Cryptosporidium* diagnosis requisition and is more sensitive than light microscopy.

Other factors may also be partially responsible for the regional disparities observed such as land use and exposure to farm animals. Two species of *Cryptosporidium* are responsible for the

majority of human infections: *C. hominis*, for which humans are the only reservoir and the transmission is human-to-human, and *C. parvum*, a zoonotic species common in both domestic animals and humans. In a study conducted in farm animals in eight provinces and one Canadian territory, *Cryptosporidium* was detected in stool samples from cattle (20%), sheep (24%), swine (11%) and horses (17%) [27]. Therefore, livestock may constitute a major source of environmental contamination, which could potentially explain some of the regional differences observed in this study, as some regions are urban and some rural. Indeed, the exposure to livestock, either through direct contact with feces or contaminated water, was used by Majowicz et al., 2001 as an explanation to the high proportion of rural cases observed in Ontario between 1996–1997 [28]. Genotyping performed on a limited number of samples in the present study, appeared to support this hypothesis, as close to 75% of human cases genotyped were of the *C. parvum* species. Moreover, subtyping analyses revealed that the majority of these were of the *C. parvum* IIaA15G2R1, *C. parvum* IIaA16G3R1 and *C. parvum* IIaA16G2R1 subtypes. In the United-States and Canada, as well as in other countries such as England, Portugal, Australia, Japan and Kuwait, the *C. parvum* IIaA15G2R1 subtype is the most commonly identified zoonotic infection in calves as well as in humans [29]. A Canadian study also reported the same three subtypes in cattle [30]. Interestingly, the data acquired through the epidemiological investigation confirmed that a third of cases had contact with livestock during their exposure period. In the largest study to date of sources of contamination with *Cryptosporidium* in outbreak cases in the UK, *C. parvum* was associated in 42% of outbreaks and mainly due to farm animal contact [25].

A bimodal distribution for age groups was observed for the reported cases, with a first peak for children younger than five years of age and a second peak for adults between 20 to 30 years of age. The age distribution seen in this study is similar to those of surveillance reports from the United States [5,31] and Europe [32] and is often associated with human-to-human transmission from a young child to a close caregiver [5].

Females represented only a slightly larger proportion of reported cases (55%) than males (45%). However, when looking at the average annual crude incidence rate by age group for both sexes, females appear to have higher crude incidence rates than males for groups between the ages of 20 to 34 but lower for groups between the ages of 5 to 14. This age difference between the sexes has been reported in the literature, where it has been explained by the fact that females older than 15 years are more likely to play a role of caretaker to young children, increasing their risk of infection [33]. It is also possible that males are less likely to consult a health practitioner in the event of symptoms and, thus, are less likely to have laboratory confirmed results [31,32]. The high proportion of hospitalizations among investigated cases (15%) is worth mentioning and needs to be further explored. During the same time period, hospitalization rates in the neighbouring province of Ontario were 5.1% and 3.9% for 2016 and 2017, respectively [34]. However, this may also be the result of selection bias as more severe cases may be hospitalized and tested for confirmation of an etiological agent.

One of the limitations of the present study concerns the data resulting from the epidemiological investigation. The investigation was done on a voluntary basis by RPHD. In fact, only 36% of all reported cases during 2016 and 2017 were investigated. Given the fact that the present study does not have a comparison group, the risk factors associated with the acquisition of cryptosporidiosis in Québec were not determined. Only the data on exposures were analysed descriptively.

Another limitation is the low number of specimens collected for genotyping and subtyping analysis. In Québec, the most common method used for the detection of *Cryptosporidium* in stool specimens involves microscopy using modified acid-fast staining. This method requires that the stool specimen be fixed in SAF. However, the extraction of DNA needed for PCR testing and molecular characterization of specimens requires the availability of fresh, unfixed

stool. Therefore, a second sample had to be rapidly obtained from patients for genotyping and subtyping purposes. The difficulty involved in obtaining a second sample from patients resulted in a reduction in the number of participating laboratories, specifically for those that use fixed samples for the diagnosis of cryptosporidiosis. For laboratories that already used PCR as the diagnostic tool, a second sample was not required and the original frozen sample could be sent directly to the LSPQ. Therefore, the majority of samples analysed in this study came from laboratories that use PCR as the detection method.

A third limitation is related directly to the underdiagnosis of the parasite. The current method used for the detection of parasites in most medical microbiology laboratories of Québec, stool microscopy using modified acid-fast staining in specimens, does not allow for the detection of *Cryptosporidium* oocysts. This leads to the underdiagnosis of the disease as the detection of *Cryptosporidium* then requires an additional step where doctors have to explicitly request a search for the parasite in addition to the standard parasitological test. Taken together with the fact that many people will be asymptomatic [2] the reported cases in the NIDR represent the more severe cases of the disease.

The differences seen in the adjusted incidence rate between the HRR may be related in part to a detection bias associated to the laboratory services offered in each region. Investigated cases cannot be considered representative of reported cases; however demographic characteristics were similar between the two groups.

Finally, a last but important limitation of the present study is related to the proportion of missing data for certain variables for both reported cases in the registry and investigated cases. As it concerns the data from the NIDR, this can partly be explained by the fact that cryptosporidiosis cases are not routinely investigated by RPHD. On the other hand, for the epidemiological investigation, the latter is a telephone interview performed by an RPHD agent, such as a nurse or doctor, and is often done days to weeks after the initial declaration. This can lead not only towards memory loss for cases but also a refusal to participate in the interview.

The data obtained through the present study shows that combining genotype and subtype information with epidemiological data may be an approach to determine the contamination source and guide prevention efforts. Unfortunately, due to the lack of standard diagnostic methods and epidemiological data, a more complete portrait of cryptosporidiosis in Québec was not achieved in this study.

## Conclusions

This first brief epidemiological portrait of cryptosporidiosis in Québec has allowed for the description, both at the provincial and regional level, of the populations that could be particularly vulnerable to the disease. It has also described potential exposures recognized in the literature. The genotyping and subtyping of a limited number of samples has suggested that the dominant cryptosporidiosis species in Québec is *C. parvum*, which suggests that the transmission may be largely zoonotic.

## Acknowledgments

We thank the many health professionals of the regional public health departments involved in the investigation of the cases during the study period.

## Author Contributions

**Conceptualization:** Juliana Ayres Hutter, Réjean Dion, Marc Fiset, Julien Trépanier, Karine Thivierge.

**Data curation:** Juliana Ayres Hutter, Réjean Dion, Karine Thivierge.

**Formal analysis:** Juliana Ayres Hutter, Rebecca Guy, Jeannie Lisette Aguilar, Karine Thivierge.

**Funding acquisition:** Karine Thivierge.

**Methodology:** Juliana Ayres Hutter, Réjean Dion, Alejandra Irace-Cima, Marc Fiset, Rebecca Guy, Brent Dixon, Jeannie Lisette Aguilar, Julien Trépanier, Karine Thivierge.

**Project administration:** Marc Fiset, Karine Thivierge.

**Resources:** Juliana Ayres Hutter, Alejandra Irace-Cima, Karine Thivierge.

**Supervision:** Alejandra Irace-Cima.

**Validation:** Juliana Ayres Hutter, Rebecca Guy, Brent Dixon, Karine Thivierge.

**Writing – original draft:** Juliana Ayres Hutter.

**Writing – review & editing:** Juliana Ayres Hutter, Réjean Dion, Alejandra Irace-Cima, Marc Fiset, Rebecca Guy, Brent Dixon, Jeannie Lisette Aguilar, Julien Trépanier, Karine Thivierge.

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
