## [Decision Letter · Decision Letter 0]

29 Jan 2020

Cryptosporidium spp.: human incidence, molecular characterization and associated exposures in Québec, Canada (2016-2017)

PONE-D-19-28824

Dear Dr. Thivierge,

We are pleased to inform you that your manuscript has been judged scientifically suitable for publication and will be formally accepted for publication once it complies with all outstanding technical requirements.

With kind regards,

Guido Favia, Ph.D.

Academic Editor

PLOS ONE

Journal Requirements:

1. Please state in your methods section, whether the data you used were anonymized when you received them.

Reviewers' comments:

Reviewer's Responses to Questions

**Comments to the Author**

1. Is the manuscript technically sound, and do the data support the conclusions?

Reviewer #1: Yes

Reviewer #2: Yes

2. Has the statistical analysis been performed appropriately and rigorously? 

Reviewer #1: Yes

Reviewer #2: Yes

3. Have the authors made all data underlying the findings in their manuscript fully available?

Reviewer #1: Yes

Reviewer #2: Yes

4. Is the manuscript presented in an intelligible fashion and written in standard English?

Reviewer #1: Yes

Reviewer #2: Yes

5. Review Comments to the Author

Reviewer #1: The manuscript: Cryptosporidium spp.: human incidence, molecular characterization and associated exposures in Québec, Canada (2016-2017) is well written, clear, and easy to understand and in my opinion should be published in the selected journal. A large amount of work was involved in the study in identifying Cryptosporidium infections in human. The epidemiology of human cyptosporidiosis presents unique problem encompassing the need to clarify possible ways of infections and their zoonotic potential. These data will have a contribution to understand the transmission of Cryptosporidium in Québec.

Reviewer #2: In this article is very well described the Cryptosporidium epidemiology in Quebec from 2016 to 2017. Moreover, the article provide information about the Cryptosporidium species responsible of the disease and despite all the limitations related to the low samples analyzed it gives a good basis for further investigations.

6. PLOS authors have the option to publish the peer review history of their article (what does this mean?). If published, this will include your full peer review and any attached files.

Reviewer #1: No

Reviewer #2: No

---

## [Editor Report · Acceptance letter]

3 Feb 2020

PONE-D-19-28824 

Cryptosporidium spp.: human incidence, molecular characterization and associated exposures in Québec, Canada (2016-2017) 

Dear Dr. Thivierge:

I am pleased to inform you that your manuscript has been deemed suitable for publication in PLOS ONE. Congratulations! Your manuscript is now with our production department. 

With kind regards,

on behalf of

Prof. Guido Favia 

Academic Editor

PLOS ONE